# Does the Ketogenic Diet Mediate Inflammation Markers in Obese and Overweight Adults? A Systematic Review and Meta-Analysis of Randomized Clinical Trials

**DOI:** 10.3390/nu16234002

**Published:** 2024-11-22

**Authors:** Mariangela Rondanelli, Clara Gasparri, Martina Pirola, Gaetan Claude Barrile, Alessia Moroni, Ignacio Sajoux, Simone Perna

**Affiliations:** 1Department of Public Health, Experimental and Forensic Medicine, University of Pavia, 27100 Pavia, Italy; mariangela.rondanelli@unipv.it; 2Endocrinology and Nutrition Unit, Azienda di Servizi alla Persona ‘‘Istituto Santa Margherita’’, University of Pavia, 27100 Pavia, Italy; martina.pirola03@universitadipavia.it (M.P.); gaetanclaude.barrile01@universitadipavia.it (G.C.B.); alessia.moroni02@universitadipavia.it (A.M.); 3Scientific Officer, PronoKal Group, 08009 Barcelona, Spain; ignacio.s@pronokal.com; 4Division of Human Nutrition, Department of Food, Environmental and Nutritional Sciences (DeFENS), Università degli Studi di Milano, 20122 Milan, Italy

**Keywords:** ketogenic diet, inflammation, obesity, ketones, CRP

## Abstract

**Background/Objectives.** The ketogenic diet has emerged as a potential treatment strategy for reducing inflammation. The purpose of this meta-analysis and systematic review is to look into how a ketogenic diet affects inflammatory biomarkers in persons who are overweight or obese. **Methods**. We conducted an extensive search of Web of Science, PubMed, Scopus, and Google Scholar to find pertinent studies reporting changes in inflammatory biomarkers such as C-reactive protein (CRP), the erythrocyte sedimentation rate, and cytokines after a ketogenic diet. **Results.** Seven randomized controlled trials involving 218 overweight or obese individuals who followed a ketogenic or control diet over 8 weeks to 2 years were included in the review, and five of those were considered for the meta-analysis. The primary outcomes were CRP and IL-6 levels. The results reported significant decreases after treatment for CRP (mean of −0.62 mg/dL (95% CI: −0.84, −0,40), and a slight, but not statistically significant, reduction in IL-6 (mean of −1.31 pg/mL (95% CI: −2.86, 0.25). **Conclusions**. The ketogenic diet could contribute to modulating inflammation in obese and overweight subjects.

## 1. Introduction

Obesity is often accompanied by chronic low-grade inflammation, which is associated with metabolic diseases and organ tissue complications, such as insulin resistance, glucose intolerance, and type 2 diabetes [1,2]. The complex interconnections between obesity and inflammation are mainly due to the intricate cross-talk between various pro- and anti-inflammatory pathways within expanding fat stores, particularly in visceral adipose tissue (VAT) [3]. Adipose tissue is an active endocrine organ that secretes various molecules called adipokines, able to modulate inflammatory responses, regulating the production of pro- and anti-inflammatory cytokines by immune cells [4].

Although the precise causes of obesity-related inflammation are unclear and may differ across tissues, it is commonly known that abnormalities in adipokine synthesis by adipose tissue caused by excess visceral fat result in persistent low-grade inflammation [5]. The association between chronic inflammation and obesity-associated metabolic disturbances is now widely recognized, prompting numerous studies to investigate the specific biomarkers of oxidative stress and systemic inflammation in this context.

Visceral fat has a pivotal role in metabolic disturbances, and various adipokines and pro-inflammatory cytokines secreted by visceral adipocytes may be involved in altered metabolism [6]. Inflammation plays a crucial role in the pathogenesis of metabolic syndrome in obese and overweight individuals, and adipose tissue dysregulation and increased monocyte/macrophage activity appear to be key drivers of this inflammatory process. In a recent study, Reddy et al. [7] identified elevated that C-reactive protein (CRP) levels are a prototypic biomarker of the inflammatory burden in metabolic syndrome.

Since it was found in 1921 that both starvation and high-fat diets cause ketosis, the ketogenic diet (KD) has been used to treat epilepsy. The ketogenic diet acquired popularity as a possible treatment for several illnesses towards the end of the 20th century [8]. Low-carbohydrate and ketogenic diets have been linked to beneficial effects on inflammatory biomarkers in individual studies, but this association has never been systematically reviewed [8]. The ketogenic diet is a low-carbohydrate, high-fat, and adequate-protein diet [9]. To induce nutritional ketosis, which is defined as an elevation of ketones in the blood in the range of 0.5–3.0 mmol/L, the ketogenic diet reduces carbohydrate intake to below 50 g a day [10].

It has been demonstrated that ketones can control oxidative stress and inflammation [11]. In particular, β-hydroxybutyrate (βOHB) inhibits the NLRP3 inflammasome and activates the GPR109A receptor to reduce inflammation in macrophages. The βOHB-mediated inhibition of this inflammasome reduces proinflammatory interleukin 1β (IL-1β), IL-18, and caspase-1 activation, independently of GPR109A [12].

The classic ketogenic diet is the most widely evaluated KD, although other forms of this diet, such as the medium-chain triglyceride diet, the modified Atkins diet, low glycemic index treatment, and the very low-calorie ketogenic diet (VLCKD), present several advantages due to their better tolerability and easier administration [8].

When compared to low-calorie diets, ketogenic dietary therapy has been linked to a higher reduction in VAT and a stronger preservation of lean mass [13]. By lowering the factors that cause the release of proinflammatory cytokines, a decrease in VAT can result in less chronic inflammation. Following a VLCKD intervention, one study found that the levels of serum CRP decreased and anti-inflammatory cytokines increased [14].

CRP is generated from the liver in response to inflammation and has a positive correlation with visceral fat, which is commonly seen in obese people, and it is a risk factor for cardiovascular events and type 2 diabetes [15].

Given the foregoing, the purpose of this systematic review and meta-analysis was to examine the effects of a ketogenic diet on cytokine-based indicators of inflammation in overweight or obese people, as opposed to other dietary regimens.

## 2. Materials and Methods

### 2.1. Eligibility Criteria

Eligible studies were English-language randomized controlled trials (RCTs) published in the previous 10 years that examined the effects of the ketogenic diet on inflammation in adults. The search targeted studies reporting on serum CRP, serum interleukin, TNF-α, albumin, insulin, glucose, total cholesterol, HDL, LDL, and triglycerides. In order to calculate mean changes, the relative standard deviations from baseline, and/or the mean differences between intervention and control groups, the eligible studies had to supply enough data. Studies involving children, studies that were not randomized, and studies without a control group were not eligible for inclusion.

This review was conducted in four steps, guided by the Preferred Reporting Items for Systematic Review and Meta-Analyses (PRISMA) statement [16]: (1) the formulation of the review question: “Can the ketogenic diet affect the inflammatory process?”; (2) the definition of participants: overweight/obese women and men aged 18 to 75 years; (3) the approach used to search for pertinent intervention studies that discussed how the ketogenic diet affected the inflammatory process; and (4) data analysis using a meta-analysis and systematic review.

### 2.2. Information Sources

A search of PubMed, Scopus, Web of Science, and Google Scholar for English-language articles published between January 2013 and July 2024 was performed.

### 2.3. Search Strategy

The following search terms were employed: ketogenic diet [MeSH Terms]) OR very low-calorie ketogenic diet [MeSH Terms]) OR ketosis [MeSH Terms]) OR protein diet [MeSH Terms]) OR low-carbohydrate [MeSH Terms]) OR low-carbohydrate diet [MeSH Terms]) OR carbohydrate-restricted [MeSH Terms]) OR nutritional ketosis [MeSH Terms]) OR high-fat [MeSH Terms]) AND inflammation [MeSH Terms]) OR cytokines [MeSH Terms]) OR inflammatory diseases [MeSH Terms]) OR C-reactive protein [MeSH Terms]) OR lipid profile [MeSH Terms]) OR visceral adipose tissue [MeSH Terms]) OR lipids [MeSH Terms]).

### 2.4. Study Selection

Studies retrieved by the search strategy were screened and selected for a full-text review, performed independently by two authors (C.G. and M.P.), based on the predefined inclusion and exclusion criteria. In the case of disagreement, a third author was involved in the decision process (A.M.).

### 2.5. Participants

The eligibility criteria for participants included an age of 18 years or older and being overweight or obese, defined as a body mass index (BMI) of 25 kg/m^2^ or greater, and treatment with a ketogenic or control diet. No restrictions were placed on gender, diseases, race, or geographic location.

### 2.6. Intervention

RCTs investigating the effects of the ketogenic diet on inflammatory biomarkers in overweight and obese individuals were included. The range of anthropometric and biochemical markers was wide and included serum levels of adiponectin, TNF-α, IL-10, IL-6, albumin, insulin, CRP, and glucose. Levels reported in different units were standardized for the meta-analysis.

### 2.7. Risk of Bias in Individual Studies

The Cochrane Collaboration Risk of Bias tool was used to evaluate the risk of bias in individual studies [17]. The creation of the allocation sequence, allocation concealment, blinding of outcome data, existence of incomplete data, and selective reporting were among the factors evaluated to determine the quality of the study. In each case, the risk was classified as low, high, or unclear. Studies with a low risk of bias for at least three items were considered to be of high quality, those with a low risk of bias for at least two items were considered to be of fair quality, and those with a low risk of bias for one or no items were considered to be of poor quality. This was evaluated independently by two writers (M.P. and C.G.), and any discrepancies were settled by a third author (S.P.).

### 2.8. Data Extraction and Analysis

Two authors (S.P. and M.R.) independently extracted the data and recorded the following for each study: first author and publication year; study design; study setting; inclusion criteria; number, gender/sex, and age of trial participants; dietary intervention in the control and experimental group(s); duration of interventions; and primary outcomes observed in each group. A meta-analysis was conducted to provide a pooled estimate for aggregated data.

### 2.9. Statistical Analysis

To collect ambiguous or missing data, the study’s authors were contacted. Missing data for continuous outcome data were taken into account using the same techniques as in the original research (usually last observation carried forward or mixed model repeated measures). Using *p* values, missing SD values were calculated. The meta-analysis combined results using the pooled effect size, which was the standardized mean difference with 95% CI. Heterogeneity across the studies was assessed using the *I*^2^ statistic. A fixed-effects model for data pooling was used when this statistic was below 50%. An *I*^2^ of less than 50% in Higgins’ *I*^2^ statistic test and a non-significant result in Cochrane’s Q test for significance indicate acceptable heterogeneity among studies. A meta-analysis or subgroup analysis of datasets from five or more research was used to investigate publication bias. Procedures related to data pooling were carried out in Review Manager 5.4.

## 3. Results

The search strategy retrieved 324 studies. Of these, 282 were removed after initial screening and the removal of duplicates. Of the 42 remaining studies, 30 were eliminated as they were not RCTs. This left 12 studies for the full-text review. Five were eliminated due to methodological issues (they did not assess dietary intake during treatment). The meta-analysis and systematic review covered the remaining seven. Figure 1 depicts the flowchart of the study selection.

Most of the studies analyzed a classic ketogenic diet with a total daily energy intake divided as follows: 15% carbohydrates, 60% lipids, and 25% proteins. The ketogenic dietary intervention was mostly compared with a baseline diet consisting of 15% proteins, 50% carbohydrates, and 35% lipids or the usual diet followed by the participants.

Although the initial study design contemplated a wide range of biomarkers, TNF-α, IL-10, albumin, insulin, glucose, total cholesterol, HDL, LDL, and triglycerides were not included in the meta-analysis as fewer than three studies analyzing each of these measures were yielded by the literature search.

The main features of the seven RCTs included in the systematic review are summarized in Table 1. The participants’ ages ranged from 18 to 75 years. The total number of participants was 218. In five of the studies, the participants in the intervention group followed a ketogenic diet, while those in the control group followed a whole-food diet (without ultra-processed foods) or their usual diet [18,19,20,21,22]. In two studies, the intervention was a VLCKD with or without docosahexaenoic acid (DHA) supplements [23,24]. Generally, the classic ketogenic diet was formulated with 90% lipids, 6% protein, and 4% carbohydrates, with a very high ketogenic ratio of 4:1. The VLCKD used had a total calorie intake of 600–800 kcal/d, a carbohydrate intake of 20–60 g/d, obtained from plant foods, 1.2–1.5 g/kg of protein per kilogram of ideal body weight from food sources with high biological value to preserve muscle mass, and 15–30 g of lipid intake obtained mainly from extra virgin olive oil and omega-3 series polyunsaturated fatty acids. In one study, a medium-carbohydrate, low-fat, calorie-restricted, carbohydrate-counting diet or a very low-carbohydrate, high-fat, non-calorie-restricted diet were administered to induce nutritional ketosis [21].

Overall, almost all of the studies considered in this review showed improvements in inflammatory status, specifically with reductions in CRP and IL-6 levels [22,23,24,25,26].

### 3.1. Meta-Analysis

CRP and IL-6 levels were assessed through blood tests and reported as mg/L, mg/dL, or mmol/L (CRP) or pg/mL or ng/mL (IL-6). For the meta-analysis, the units were standardized to mg/dL and pg/mL, respectively. The effects of the ketogenic diet on CRP levels across four studies (references) are depicted in Figure 2, which shows a significant decrease after treatment (mean of −0.62 mg/dL (95% CI: −0.84, −0,40)).

Figure 3 illustrates the effects of the ketogenic diet on IL-6 levels over time. In this case, a not statistically significant decrease was detected, although the analyzed studies were only two (mean of −1.31 pg/mL (95% CI: −2.86, 0.25)).

### 3.2. Risk of Bias

The risk of bias for studies included in the meta-analysis, according to the Cochrane Risk of Bias Tool, is reported in Table 2.

## 4. Discussion

This systematic review and meta-analysis showed evidence that the ketogenic diet could be used as a treatment for lowering inflammatory biomarkers in obese and overweight adults. We analyzed seven RCTs involving 288 participants who followed a ketogenic or control diet for 8 weeks to 2 years. The meta-analysis showed statistically significant changes in the CRP levels after treatment in mostly overweight or obese adults; a slight, but not statistically significant, reduction in IL-6 levels was observed. There were insufficient data to analyze other inflammatory markers, such as TNF-α and the erythrocyte sedimentation rate.

It has long been known that diet and fasting have a mitigating effect on inflammation, but the precise mechanisms by which this occurs remain largely elusive. The ketone bodies β-hydroxybutyric acid (BHB) and acetoacetate (AcAc) play a crucial role in mammalian survival during energy-deficit states, as they provide an alternative source of adenosine triphosphate. BHB levels are high during the ketogenic diet. BHB inhibits NLRP3, an inflammasome complex involved in driving inflammatory responses in many diseases, including several autoinflammatory conditions; it inhibits inflammatory activity by preventing the efflux of potassium ions, therefore decreasing IL-1 and IL-18 production [27]. Weight loss, a decrease in VAT, the modification of inflammatory markers, and the treatment of chronic pain with a resultant improvement in quality of life are some of the potential advantages of the ketogenic diet that have been documented [19]. The anti-inflammatory effects of the ketogenic diet are possibly mediated by several pathways. One recent study showed that this dietary therapy induced the cytochrome P450 4 A-dependent ω- and ω-1-hydroxylation of reactive lipid species, a novel mechanism that might contribute to the anti-inflammatory properties observed with ketogenic diet therapy [28]. The ketogenic diet also reduces the production of reactive oxygen species, thereby helping to improve mitochondrial respiration and bypass complex I dysfunction [29]. As suggested in a recent review of dietary therapy in obese individuals, there may be more specific mechanisms through which the ketogenic diet improves low-grade chronic inflammation and oxidative stress [30]. Ketosis specifically (1) inhibits NLRP3, a crucial signaling platform that triggers pro-inflammatory cytokines like IL-1β and IL-18; (2) raises adenosine levels, which enhances HIF-α activation; (3) activates GPR109A, a receptor that has been shown to have anti-inflammatory properties; and (4) is linked to weight loss and calorie restriction, which alter the expression of pro- and anti-inflammatory genes [30].

The findings of a recent study suggest that DHA supplementation may enhance the anti-inflammatory effects of the KD [23]. The authors compared a VLCKD with and without DHA supplementation. Large volumes of free fatty acids and bioactive lipid mediators, which are made from fatty acids and have potent pro- and anti-inflammatory effects, are released by adipose tissue during a VLCKD. Although the diet alone led to an improved red blood cell fatty acid profile (increased omega-3 index and decreased ratios of arachidonic acid [AA] to eicosapentaenoic acid [EPA] and AA to DHA), the anti-inflammatory fatty acid index (AIFAI) fell significantly during the ketogenic phase but reverted to near-baseline levels during the re-education phase. When the diet was combined with DHA supplementation, the AIFAI increased during both the ketogenic and re-education phases. DHA supplementation was also associated with greater variability in the omega-3 index and AA/EPA and AA/DHA ratios. Research has shown that DHA supplementation induces the time-dependent incorporation of omega-3 polyunsaturated fatty acids into cell membrane phospholipids. This could affect inflammation as it would result in lower AA availability for eicosanoid synthesis [23]. Indeed, it is well known that taking DHA supplements raises the levels of intermediate anti-inflammatory metabolites, which in turn affects the release of proinflammatory substances by increasing resolvine and other active metabolites to address lipo-inflammation. As a result, even over an extended period of time, DHA administration is able to disrupt the re-entry circuit between obesity and lipo-inflammation, resulting in a decreased risk of weight gain and an increased sense of satiety [31,32,33].

Despite the limited number of studies analyzed, especially concerning IL(6), an overall significant decrease in CRP and IL(6) was found, indicating the promising modulating effect of the ketogenic diet, which could also be linked to a potential effect on the expression of genes related to enzymes involved in the synthesis of pro-inflammatory (ALOX5, COX1, COX2) and anti-inflammatory (ALOX15) eicosanoids. Based on the results of a study of gene expression in multiple sclerosis, Bock et al. [34] suggested that ketogenic diets might counteract pro-inflammatory mediators such as leukotrienes by increasing vascular permeability, promoting leukocyte migration, and improving leukocyte chemotaxis.

Adipose tissues are among the several organs that secrete the proinflammatory cytokines CRP and IL-6. These cytokines are more prevalent in obese individuals than in non-obese individuals, and there is a positive correlation between obesity and fat mass. Long-term exposure to elevated levels of CRP and IL-6 is associated with type 2 diabetes, insulin resistance, and metabolic syndrome [35]. Subclinical, chronic inflammation is believed to originate in adipose tissue and may play a role in the development of metabolic syndrome. More precisely, the development of visceral adipose tissue is believed to be the primary source of the elevated inflammation associated with obesity. During this phase, macrophage infiltration in adipose tissue results in chronic oxidative stress and an inflammatory response due to altered immunological responses and endothelial dysfunction. These factors are underlying causes of diabetes, cardiovascular diseases, and other metabolic abnormalities [36]. Therefore, reducing the level of inflammation could reduce the risk of developing these disorders.

Regarding IL-6, the results of the present study are in line with those reported in an interesting very recent metanalysis that included randomized controlled trials investigating the effects of a KD on CRP, tumor necrosis factor-alpha (TNF-α), IL-6, IL-8, and IL-10 levels, showing that this KD has an effect on lowering TNF-a and IL-6 [37]. However, the results regarding CRP are not entirely similar; in fact, unlike the study conducted by Ji et al. [37], the results of the present meta-analysis showed a statistically significant reduction in CRP. The quantitative analysis regarding TNF-α was not conducted due to a lack of studies deemed adequate for the purpose, having restricted the search only to studies published in the last 10 years.

One of the key strengths of this meta-analysis lies in the design of the studies reviewed, as they were all RCTs, which enable causative inferences to be drawn. The examination of CRP and IL-6 levels before and after the various dietary interventions analyzed lends strength to the robustness of our findings. This systematic review and meta-analysis also has some limitations, including the small number of studies eligible for inclusion and the heterogeneous patient profiles. Most of the trial participants were overweight or obese, and some of them had comorbid conditions; moreover, it would have been interesting to consider other inflammatory indices, such as TNF-alpha, but the studies currently published on the subject did not fit the criteria we had defined for this meta-analysis.

## 5. Conclusions

This meta-analysis showed the potential ability of the ketogenic diet to lower inflammatory biomarkers in overweight and obese individuals through different mechanisms. Such results need to be improved in order to involve a wider number of subjects and consider other inflammatory markers that could not be analyzed in this study. For this reason, these findings could be a starting point for future studies that investigate the long-term effectiveness and safety of this diet in achieving sustained reductions in inflammation levels.

## Figures and Tables

**Figure 1 nutrients-16-04002-f001:**
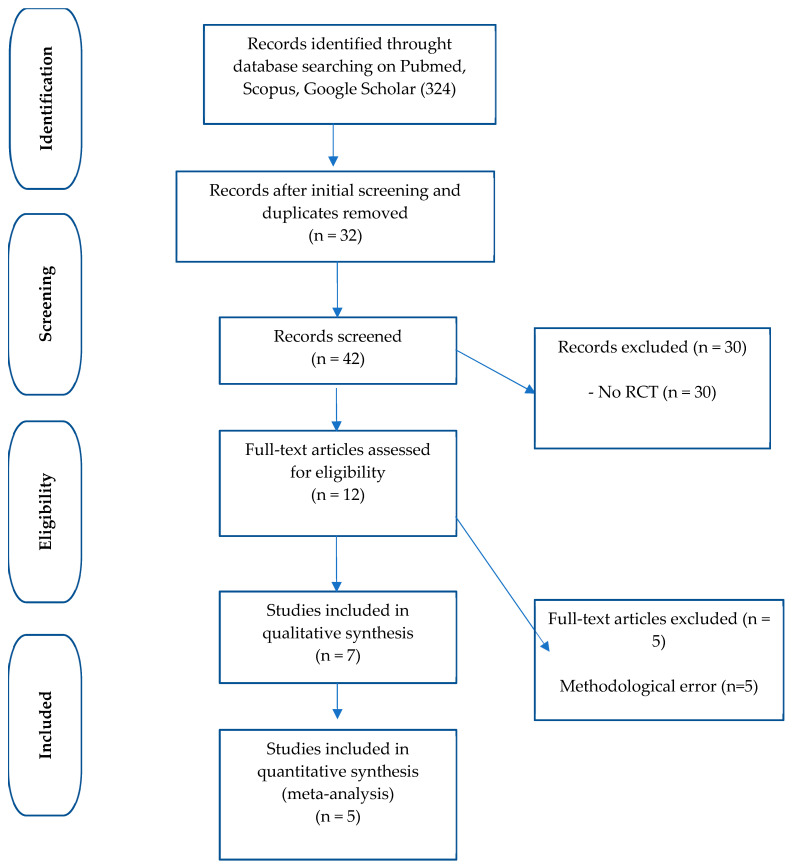
Flow diagram of the study.

**Figure 2 nutrients-16-04002-f002:**
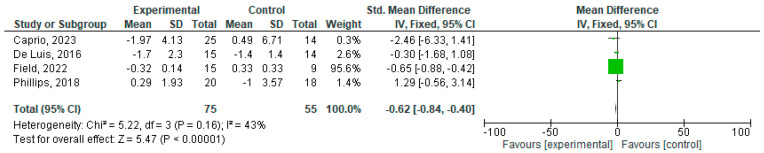
Effects of KD compared other type of diets on CRP [18,19,23,24].

**Figure 3 nutrients-16-04002-f003:**
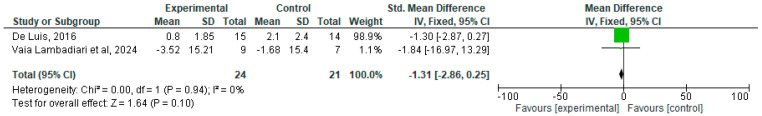
Effects of KD compared to other types of diet on IL6 [22,23].

**Table 1 nutrients-16-04002-t001:** Studies included in the meta-analysis.

First Author, Year	Study Design	Participants	Treatment	Parallel Treatments	Duration	Outcomes Related to Inflammatory Status	Results
Rosenbaum, 2019 [20]	RCT	17 men without diabetes and with BMI between 25–35 kg/m^2^	Isocaloric 15% protein, 5% carbohydrate, 80% fat ketogenic diet (KD) on energy expenditure	/	8 weeks	Cytokines, Inflammatory Markers	CRP were significantly increased on the KD.
Phillips, 2018 [18]	RCT	38 patients aged 40 to 75 years with a diagnosis fulfilling the UK Parkinson’s Disease Society Brain Bank criteria, MoCA score > 20, able and willing to follow either diet plan.	Each caloriebooster recipe added an average of 500 extra kcal, which was made up of 50 g of fat (22 g saturated), 6 g of protein, 5 g of net carbohydrate, and 4 g of fiber. The ketogenic plan provided 1750 kcal per day, which was made up of 152 g of fat (67 g saturated), 75 g of protein, 16 g net carbohydrate, and 11 g of fiber.	For those with higher energy needs, ad libitum “caloriebooster” recipes each provided an average of 500 extra kcal, consisting of 4 g of fat (1 g saturated), 6 g of protein, 102 g of net carbohydrate, and 13 g of fiber. The low-fat plan provided 1750 kcal per day, which was made up of 42 g of fat (10 g saturated), 75 g of protein, 246 g net carbohydrate, and 33 g of fiber.	8 weeks	CRP	There were no variations in CRP between groups or within groups.
Field, R., 2022 [19]	RCT	27 (23 female, 4 male) The group sizes were not equal (WFD n = 9 and WFKD n = 15). Participants were included if they were 18 or over, had experienced chronic musculoskeletal pain for over three months, were currently eating a standard western diet, had baseline pain VAS of 30 mm, and agreed to the requirements of the study.	Well-formulated ketogenic diet (WFKD) with a reduction in carbohydrate intake to between 30 and 50 g/day to achieve nutritional ketosis with a ketone level of 0.5–3.0 mmol/L	whole-food diet (WFD), removing ultra-processed foods	12 weeks	CRP, ESR	The WFKD group demonstrated significant improvements in inflammation (CRP)
De Luis, D., 2016 [23]	RCT	Control group (n = 15), PnK-DHA group (n = 14); healthy individuals. The inclusion criteria were age 18–65 years, body mass index (BMI) between 30 and 35, stable body weight in the previous 3 months, and desire to lose weight.	PnK-DHA group: This group used a commercial weight loss program (the PnK Method) that prescribed a ketogenic diet. A ketogenic diet that is extremely low in calories (600–800 kcal/day), low in carbohydrates (less than 50 g daily from veggies), and low in fats (only 10 g of olive oil per day) is what the intervention stage entails. In order to ensure that the body met its minimum requirements and to prevent the loss of lean mass, the amount of high-biological-value proteins ranged between 0.8 and 1.2 g per kilogram of ideal body weight.	isocaloric VLCK diet without DHA	6 months	Proinflammatory and pro-resolving PUFA metabolite mediators (lipid mediator profile)	TNF alpha, resistin, and C-reactive protein were all significantly altered by the VLCK diets. The intervention group’s DHA-derived oxylipin levels dramatically rose after DHA supplementation. Over the course of the trial, the intervention group’s plasma showed an increase in the ratio of roresolution to proinflammatory lipid markers. The anti-inflammatory fatty acid index (AIFAI) was consistently elevated following the DHA therapy (*p* < 0.05), and the mean ratios of AA/EPA and AA/DHA in erythrocyte membranes were significantly decreased in the PnK-DHA group. The anti-inflammatory effect of a very low-calorie ketogenic diet supplemented with DHA was significantly improved.
Saslow et al., 2014 [21]	RCT	34 participants aged 18 or over with a diagnosis of type 2 diabetes mellitus or prediabetes Participants also needed to have a BMI of 25 or above	A very low-carbohydrate, high-fat, non-calorie-restricted diet whose goal was to induce nutritional ketosis (LCK, n = 16).	A medium carbohydrate, low-fat, calorie-restricted, carbohydrate-counting diet (MCCR) consistent with guidelines from the American Diabetes Association (n = 18)	3 months	CRP	CRP declined in both groups
Vaia Lambadiari et al., 2024 [22]	RCT	16 patients met the following criteria: (1) they were older than 18, (2) their BMI was greater than 30 kg/m^2^, (3) they were diagnosed with psoriasis and psoriatic arthritis, (4) they had been prescribed biologic agents and/or synthetic disease-modifying anti-rheumatic drugs (DMARDs) for at least three months, (5) their PASI score improvement was less than 75%, and (5) their joints had moderate to severe activity (>3 swollen and >3 tender joints or DAPSA < 14).	A daily calorie intake of 1550 (±50). About 34% of the KD was protein, 55% was fat, and 11% was carbohydrate.	A daily calorie intake of 1550 (±50). The MD supplied 40% fat, 40% carbs, and 20% protein.	8 weeks	Biochemical markers of inflammation (IL-6, IL-17, IL-22, IL-23)	Patients’ IL-6 levels significantly decreased after KD (*p* = 0.047) when compared to baseline, and this decrease was two times more than that of MD (−55.6% vs. −20.56%, *p* = 0.041). When compared to baseline, there was no discernible change in IL-6 following MD (*p* = 0.666). When compared to baseline, KD significantly decreased IL-17 (*p* = 0.042) and IL-23 (*p* = 0.037), while MD showed no discernible changes in the aforementioned markers (*p* > 0.05). All patients showed no change in IL-22 (*p* > 0.05).
Caprio et al., 2023 [24]	RCT	57 patients were enrolled in the study and randomly assigned to VLCKD (n = 29) or HBD (n = 28). 27 kg/m^2^ < BMI < 35 kg/m^2^ Diagnosis of HFEM Signing of the informed consent Migraine onset < 50 years Preventive migraine treatment discontinuation since at least 3 months (including RAAS inhibitors)	VLCKD diets prepared by New Penta s.r.l. Diet provided 75–105 g protein/day, 30–50 g carbohydrate/day) and 20 g lipid/day. The amount of daily fiber intake was approximately 25 g/day, as requested from Italian Guidelines (LARN 2014), mostly deriving from vegetable servings with low glycaemic index. Total energy intake was <800 kcal/day. Recommended water intake was at least 2.5 lt/day.	Total daily average energy intake was 1500–1600 kcal/day, and macronutrient composition was based on the Mediterranean Diet [lipid: 30% of total daily energy (10% MUFA, 10% PUFA, 10% SFA); carbohydrates: 55% of total daily intake; daily protein intake was approximately 0.8–1.5 g/kg of ideal body weight].	12 weeks	inflammatory markers	Only the VLCKD group showed a substantial decrease in CRP at week 12 (*p* < 0.05), supporting the well-known anti-inflammatory impact of VLCKD.

Abbreviations: AUC, area under the curve; BMI, body mass index; CRP, C-reactive protein; DHA, docosahexaenoic acid; EPA, eicosapentaenoic acid; ESR, erythrocyte sedimentation rate; RCT, randomized controlled trial; VLCKD, very low-calorie ketogenic diet; WFD, whole-food diet; WFKD, whole-food/well-formulated ketogenic diet.

**Table 2 nutrients-16-04002-t002:** Risk of bias for studies included in the meta-analysis according to the Cochrane Risk of Bias Tool ^a^.

Study, Year (Ref)	Random-Sequence Generation	Allocation Concealment	Participant- Personnel Blinding	Outcome- Assessment Blinding	Incomplete Outcome Data	Selective Reporting	Other Bias
Caprio, 2023 [24]	Low	Unclear	Low	Unclear	Low	Low	Low
Phillips, 2022 [18]	Low	Low	Low	Unclear	Low	Low	Low
Field, 2018 [19]	Unclear	Unclear	Low	Unclear	Low	Low	Low
De Luis, 2016 [23]	Low	Low	Low	Unclear	Low	Low	Low
Lambadiari, 2014 [22]	Low	Low	Low	Unclear	Low	Low	Low

^a^ Seven domains indicate bias designations based on the study criteria: low risk if the study design’s negative aspects were unlikely to affect the study’s findings; high risk if the study design was likely to affect the study’s findings; and unclear risk if there was insufficient evidence to determine whether the risk was high or low.

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
