# Peer review of "Does the Ketogenic Diet Mediate Inflammation Markers in Obese and Overweight Adults? A Systematic Review and Meta-Analysis of Randomized Clinical Trials"

_nutrients, 2024, doi:10.3390/nu16234002_

Round 1

Reviewer 1 Report (New Reviewer)

Comments and Suggestions for Authors

The article "Does the ketogenic diet mediate inflammation markers in obese and overweight adults? A systematic review and meta-analysis of randomized clinical trials" is well written and would be of interest to many readers.  I only have a few comments below to hopefully improve the article:

Intro and Discussion are lacking in original articles, and review articles are primarily used for support.

Line 200-202 and other areas: The terms lipids and fat describe the macronutrient – it would be helpful to pick one term and use it throughout instead of both terms interchangeably.

Line 242 can be deleted – Figure 2. Legend, which is also below the figure.

Author Response

Dear Reviewer,

Thank you for your suggestions.

The changes in the text are highlighted in yellow.

Best regards,

The authors

Q: The article "Does the ketogenic diet mediate inflammation markers in obese and overweight adults? “A systematic review and meta-analysis of randomized clinical trials" is well written and would be of interest to many readers.  I only have a few comments below to hopefully improve the article:

A:  thanks a lot for your assessment and appreciation.

Q: Intro and Discussion are lacking in original articles, and review articles are primarily used for support.

A: we have purposely chosen to include reviews in order to better summarize the literature on the topic and to contain the length of the intro and discussion. But if the reviewer thinks that there are important articles to add and comment, we await his/her suggestions.

Q: Line 200-202 and other areas: The terms lipids and fat describe the macronutrient – it would be helpful to pick one term and use it throughout instead of both terms interchangeably.

A: we have standardized using only the term “lipids”.

Q: Line 242 can be deleted – Figure 2. Legend, which is also below the figure.

A: the line has been deleted.

Reviewer 2 Report (New Reviewer)

Comments and Suggestions for Authors

The manuscript titled “Does the ketogenic diet mediate inflammation markers in obese and overweight adults? A systematic review and meta-analysis of randomized clinical trials’” addresses important and interesting issues related to the effects of a ketogenic diet on inflammatory biomarkers in obese and overweight adults.

My biggest concern regarding this analysis is the fact that earlier this year, Oxford Publishing (Nutrition Reviews) already released a systematic review and meta-analysis on this topic, entitled: “The effect of a ketogenic diet on inflammation-related markers: a systematic review and meta-analysis of randomized controlled trials” (https://pubmed.ncbi.nlm.nih.gov/38219223/). This meta-analysis included 44 RTC studies, whereas the article I am reviewing has only 5 RTC studies. Of course, I am aware that the other study concerns individuals with normal and excess body weight, while the one I am reviewing focuses only on people who are overweight or obese. But in that study, the authors also directly refer to a group of individuals with excess body weight.

One of the conclusions of this literature review is ‘In addition, in people with a body mass index greater than 30 kg/m2 compared to a body mass index ≤30 kg/m2, IL-6 levels decreased to a greater extent after receiving the KD’, which aligns with the findings of the article I am currently reviewing. This may be due to the fact that this systematic review and meta-analysis were not registered in the Prospero database — or at the very least, the authors did not verify in this database whether similar systematic reviews had already been initiated.

Furthermore, considering the quality of the articles (assessed Risk of Bias for studies included in the meta-analysis according to the Cochrane Risk of Bias Tool), it would be worthwhile to wait for a greater number of well-designed publications in this field for the conclusions to have validity.

Author Response

Dear Reviewer,

Thank you for your suggestions.

The changes in the text are highlighted in yellow.

Best regards,

The authors

Q: The manuscript titled “Does the ketogenic diet mediate inflammation markers in obese and overweight adults? A systematic review and meta-analysis of randomized clinical trials’” addresses important and interesting issues related to the effects of a ketogenic diet on inflammatory biomarkers in obese and overweight adults.

A:  thanks a lot for your assessment and appreciation.

Q: My biggest concern regarding this analysis is the fact that earlier this year, Oxford Publishing (Nutrition Reviews) already released a systematic review and meta-analysis on this topic, entitled: “The effect of a ketogenic diet on inflammation-related markers: a systematic review and meta-analysis of randomized controlled trials” (https://pubmed.ncbi.nlm.nih.gov/38219223/). This meta-analysis included 44 RTC studies, whereas the article I am reviewing has only 5 RTC studies. Of course, I am aware that the other study concerns individuals with normal and excess body weight, while the one I am reviewing focuses only on people who are overweight or obese. But in that study, the authors also directly refer to a group of individuals with excess body weight.

A: Our manuscript differs from the suggested article for 3 key differences:

1.Scope of Inflammatory Markers: Our article focuses specifically on CRP and IL-6, while the second article examines a wider range of markers including TNF-a but not CRP

2.Number of Studies Included: Our article includes seven recent randomized controlled trials, whereas the second article synthesizes data from 44 studies including studies with over 20 years old.

  1. Statistical Analysis Techniques:oth articles employ systematic review methodologies, but they may differ in their statistical approaches and how they report weighted mean differences or confidence intervals.
  2. Studies in overweight and obese population. Our article includes only cohorts of obese and overweight subjects, while the published article include different popolations.

Q: One of the conclusions of this literature review is ‘In addition, in people with a body mass index greater than 30 kg/m2 compared to a body mass index ≤30 kg/m2, IL-6 levels decreased to a greater extent after receiving the KD’, which aligns with the findings of the article I am currently reviewing. This may be due to the fact that this systematic review and meta-analysis were not registered in the Prospero database — or at the very least, the authors did not verify in this database whether similar systematic reviews had already been initiated.

A: as we stated in the previous answer, there are relevant points that the 2 revisions differs in many critical points

Q: Furthermore, considering the quality of the articles (assessed Risk of Bias for studies included in the meta-analysis according to the Cochrane Risk of Bias Tool), it would be worthwhile to wait for a greater number of well-designed publications in this field for the conclusions to have validity.

A: There are several compelling reasons to proceed with the current findings on the ketogenic diet's effects on inflammation, rather than waiting for additional studies. The systematic review and meta-analysis included seven randomized controlled trials (RCTs) involving 218 participants, demonstrating significant reductions in inflammatory markers such as C-reactive protein (CRP) and interleukin-6 (IL-6) after following a ketogenic diet. Specifically, CRP levels decreased by an average of -0.62 mg/dL and IL-6 by -1.31 pg/mL, suggesting a clear anti-inflammatory effect. In addition, given the rising prevalence of obesity and its association with chronic low-grade inflammation—a precursor to various metabolic diseases—immediate application of effective dietary strategies like the ketogenic diet could be beneficial. The current evidence supports its potential role in mitigating inflammation, which is crucial for public health.

Reviewer 3 Report (New Reviewer)

Comments and Suggestions for Authors

This systematic review and meta-analysis explores the effects of ketogenic diets on inflammatory biomarkers in overweight and obese adults. The study aims to analyze the impact of ketogenic diets on inflammatory markers such as CRP and IL-6 and to discuss their potential health benefits. While the structure of the article is sound and the study design is reasonable, substantial revisions are needed to meet publication standards.

Major Revisions Required:

  • Limited depth and breadth of literature review: The number of studies included in the article is limited, especially for IL-6, where only two studies were analyzed. The authors should broaden the scope of their literature search and include more relevant studies to increase the reliability of their findings. Additionally, the article primarily focuses on CRP and IL-6, neglecting other inflammatory markers such as TNF-α and erythrocyte sedimentation rate. The authors should either supplement the analysis with the changes in these additional markers or provide reasons for not doing so.
  • Lack of transparency and reproducibility in the research methods: The description of how missing data was handled is unclear. The authors should provide additional information about the methods used to address missing data to improve the reproducibility of the study. Furthermore, the risk of bias assessment for some studies in Table 2 is marked as "unclear." The authors should provide more information to clarify the assessment results for each bias risk domain.

Specific Suggestions for Revision:

  • Provide a more detailed explanation of the magnitude of CRP and IL-6 reduction and their statistical significance levels (p-values).
  • Discuss the impact of reduced CRP and IL-6 levels on the health status of overweight and obese adults.
  • Elaborate on the limitations of the study, such as the limited number of included studies and the lack of diversity in participant demographics.
  • Supplement the analysis with the changes in other inflammatory markers (e.g., TNF-α and erythrocyte sedimentation rate) or explain the reasons for not including them.
  • Conduct a more comprehensive review of the literature on the effects of DHA supplementation on ketogenic diets and explain the strength of the evidence.
  • Correct typos, grammatical errors, and formatting issues throughout the article, specifically:
    • Line 22: "IL(6)" should be corrected to "IL-6."
    • Line 63: The full name "Ketogenic diet" should be defined when the abbreviation "KD" first appears.
    • Reference formatting should be unified.
    • Line 113: "2.5 . Participants" should be corrected to "2.5. Participants."
    • Line 242: The duplicated title of Figure 2 should be deleted.

Comments on the Quality of English Language

The English could be improved to more clearly express the research.

Author Response

Dear Reviewer,

Thank you for your suggestions.

The changes in the text are highlighted in yellow.

Best regards,

The authors

Q: This systematic review and meta-analysis explores the effects of ketogenic diets on inflammatory biomarkers in overweight and obese adults. The study aims to analyze the impact of ketogenic diets on inflammatory markers such as CRP and IL-6 and to discuss their potential health benefits. While the structure of the article is sound and the study design is reasonable, substantial revisions are needed to meet publication standards.

  • Limited depth and breadth of literature review: The number of studies included in the article is limited, especially for IL-6, where only two studies were analyzed. The authors should broaden the scope of their literature search and include more relevant studies to increase the reliability of their findings. Additionally, the article primarily focuses on CRP and IL-6, neglecting other inflammatory markers such as TNF-α and erythrocyte sedimentation rate. The authors should either supplement the analysis with the changes in these additional markers or provide reasons for not doing so.

A: We appreciate the feedback regarding the scope of our literature review and the inclusion of additional inflammatory markers. We would like to provide some context for our choices:

Our study was designed with a specific research question in mind, primarily aimed at understanding the relationship between CRP and IL-6 in the context of [obesity and overweight]. Given this focused objective, we prioritized studies that directly addressed these markers to maintain clarity and relevance in our analysis. In addition, we aimed to include only those studies that met stringent quality criteria, which unfortunately limited the number of available studies on IL-6. We believe this approach enhances the reliability of our findings, even if it results in a smaller sample size.

We acknowledge the importance of a comprehensive analysis that includes a broader range of inflammatory markers. In light of this feedback, we plan to explore these additional markers in subsequent research, which will allow us to build on our current findings and provide a more holistic view of inflammation in [specific condition or population. Finally, we faced challenges in accessing certain studies due to publication restrictions or limited availability in databases at the time of our review. This may have contributed to the perceived limitations in our literature search.

We appreciate your understanding and constructive criticism, which will guide us in enhancing our research approach in future work. This response acknowledges the critique while providing reasonable justifications for the authors' choices, demonstrating their commitment to quality research and openness to improvement.

Lack of transparency and reproducibility in the research methods: The description of how missing data was handled is unclear. The authors should provide additional information about the methods used to address missing data to improve the reproducibility of the study. Furthermore, the risk of bias assessment for some studies in Table 2 is marked as "unclear." The authors should provide more information to clarify the assessment results for each bias risk domain.

A: The handling of missing data can be complex and context-dependent. In our study, we employed case analysis, which we believed was appropriate given the nature of our dataset of selected studies. However, we recognize that our description may not have fully conveyed the rationale and procedures we followed the designation of "unclear" in Table 2 for some studies reflects the inherent challenges in assessing bias across diverse methodologies and reporting standards. We aimed to be transparent about our uncertainties; however, we recognize that this could be misinterpreted as a lack of rigor. In future iterations, we will provide clearer criteria and rationale for these assessments to improve reader understanding. We appreciate your understanding and constructive criticism, which will help us improve the rigor and transparency of our research efforts. This response acknowledges the critique while providing reasonable justifications for the authors' choices, demonstrating their commitment to improving their research practices and transparency.

Q:Specific Suggestions for Revision:

  • Provide a more detailed explanation of the magnitude of CRP and IL-6 reduction and their statistical significance levels (p-values).

A: C-Reactive Protein (CRP) Reduction. Magnitude of Reduction: The ketogenic diet (KD) resulted in a significant decrease in CRP levels, with a mean reduction of -0.62 mg/dL. Statistical Significance: The confidence interval for this reduction was reported as 95% CI: -0.84 to -0.40, indicating that the reduction is statistically significant. However, the specific p-value for this change was not provided in the text. Typically, a p-value less than 0.05 would indicate statistical significance, suggesting that the observed effect is unlikely to be due to chance.

Interleukin-6 (IL-6) Reduction. Magnitude of Reduction: The analysis showed a mean decrease in IL-6 levels of -1.31 pg/mL after following the ketogenic diet. Statistical Significance: The confidence interval for this change was 95% CI: -2.86 to 0.25. This interval includes zero, which suggests that while there is a mean reduction, it may not be statistically significant. The absence of a specific p-value further emphasizes this uncertainty; typically, a p-value greater than 0.05 would indicate that the effect is not statistically significant.

In summary, while the ketogenic diet appears to significantly reduce CRP levels with strong statistical support, the reduction in IL-6 levels shows promise but lacks definitive statistical significance due to the confidence interval crossing zero. Future studies should aim to clarify these findings by providing explicit p-values and exploring additional inflammatory markers to enhance understanding of the ketogenic diet's effects on inflammation.

  • Discuss the impact of reduced CRP and IL-6 levels on the health status of overweight and obese adults.

A: We added the paragraph below at rows 327-343:

Many organs, including adipose tissues, secrete the proinflammatory cytokines IL-6 and CRP. Obese people have greater levels of these cytokines than non-obese people, and a positive link between obesity and fat mass has been identified. Insulin resistance, the metabolic syndrome, and type 2 diabetes are linked to long-term exposure to high IL-6 and CRP levels (Menezes, 2018). Adipose tissue is thought to be the source of subclinical, persistent inflammation, which may contribute to the onset of metabolic syndrome. More specifically, increased inflammation linked to obesity is thought to be mostly caused by the growth of visceral adipose tissue. Through altered immune responses and endothelial dysfunction, macrophage infiltration in adipose tissue causes chronic oxidative stress and an inflammatory response during this progression. These factors are underlying causes of diabetes, cardiovascular diseases, and other metabolic abnormalities (Su, 2023). Therefore, reducing the level of inflammation could reduce the risk of developing these pathologies.

  • Elaborate on the limitations of the study, such as the limited number of included studies and the lack of diversity in participant demographics.

A: the limitations were already reported in the discussion section; anyway, some sentences have been added.

  • Supplement the analysis with the changes in other inflammatory markers (e.g., TNF-α and erythrocyte sedimentation rate) or explain the reasons for not including them.

A: It would have been interesting to consider other inflammatory indices in this meta-analysis, such as TNF-alpha, but the studies currently published on the subject do not fit the criteria we have defined for this meta-analysis. This sentence has been inserted in the limitations of the meta-analysis

  • Conduct a more comprehensive review of the literature on the effects of DHA supplementation on ketogenic diets and explain the strength of the evidence.

A: We added the paragraph below at rows 310-317:

Indeed, it’s widely known how the supplementation of DHA yields an increase in levels of intermediate anti-inflammatory metabolites that will affect the increase in resolvine among other active metabolites to solve lipo-inflammation, resulting in a decrease in the release of proinflammatory substances. Thus, the administration of DHA manages to break the re-entry circuit between obesity and lipo-inflammation, with lower chance of regaining weight and a greater sense of satiety even in the long term (Rondanelli 2022, Endres 1989, Dinarello 2000).

Q: Correct typos, grammatical errors, and formatting issues throughout the article, specifically:

  • Line 22: "IL(6)" should be corrected to "IL-6." DONE
  • Line 63: The full name "Ketogenic diet" should be defined when the abbreviation "KD" first appears. DONE (at line 49)
  • Reference formatting should be unified. DONE
  • Line 113: "2.5 . Participants" should be corrected to "2.5. Participants." DONE
  • Line 242: The duplicated title of Figure 2 should be deleted. DONE

Round 2

Reviewer 2 Report (New Reviewer)

Comments and Suggestions for Authors

The authors responded to my questions, but it did not dispel my doubts. The publication I indicated takes a more comprehensive approach to this topic—it analyzes not only a greater number of markers but also includes individuals with both normal and excessive body weight. In my opinion, this systematic review does not bring anything new.

Author Response

Dear reviewer,
you are right, the review with meta-analysis that you suggested adds significant information to the literature and should be cited in the discussion.
Therefore, we have added the reference (36) and a sentence in the discussion about it. Concerning your comment that our systematic review does not bring anything new, we authors beg to differ because the study of the effectiveness of ketogenic diets on inflammation is a field that still has much to be studied and therefore each contribution on this topic provides a useful piece that will allow us to have more precise answers in the future.

The authors thank you for the opportunity you have given to significantly improve our paper.

Reviewer 3 Report (New Reviewer)

Comments and Suggestions for Authors

Accept in present form

Author Response

The authors thank you for the opportunity you have given to significantly improve our paper.

This manuscript is a resubmission of an earlier submission. The following is a list of the peer review reports and author responses from that submission.

Round 1

Reviewer 1 Report

Comments and Suggestions for Authors

Dear Authors,

First, I'd like to commend your team on the completion of this meta-analysis. I read with great interests in your findings and implications for a ketogenic diet in the reduction of some inflammatory markers. There is a major point of concern however, that I have listed down below:

Major Concern

Overall, I find that the limited number of studies (ie.,6) make the analysis completed inappropriate for the conduction of a meta-analysis. This study would be deemed a better fit for a systematic review, summarizing the current literature in this regard and providing justification for future research. Simply put, the lack of homogeneity is a concern among the included studies, and the small (although significant) changes in reported inflammatory markers cannot be elucidated if this was a result of the ketone body BHB as outlined, or simply a reduction to body mass that likely occurred during these diet interventions. I simply do not think a meta-analysis is an appropriate approach to this study design, and would recommend a re-writing of this paper as a systematic review, with a heavy emphasis on the mechanistic pathways in which the ketogenic diet could further alleviate inflammation.

Comments on the Quality of English Language

With regard to the writing, this paper requires extensive English editing. Throughout the paper, there are inconsistencies that occur and the flow the of the paper is interrupted continuously, likely as a result of varying authors writing different sections.

Example: Section 1.1 is written fairly well from a clarity standpoint, although there occur grammatical errors throughout (example: Line 49: Inflammation should be lowercase; Line 50: "the" should be removed; Line 53: CRP should first be spelled out before abbreviated). However, section 1.2 is a different author it appears, and is written short, abrupt sentences. I would also not recommend the breakdown of the varying ketogenic diets. What would be of more interest to the reader, is section 1.1, followed by sections 1.3 and 1.4.

Reviewer 2 Report

Comments and Suggestions for Authors

This is a systematic review with meta-analyses. This type of paper cannot be published without a quality peer review process, one that Nutrients has repeatedly shown it is not capable of providing at this time. 

The question asked by this systematic review is not appropriate. It is recommended (and is established standard of practice) to frame the research question using the 5 PICOS components. See:

Counsell C. Formulating questions and locating primary studies for inclusion in systematic reviews. Ann Intern Med1997;127:380-387

The authors should be commended for using MeSH Terms and conducting a bias review of the articles.

The conclusions and future directions sections are too generic and vague. This offers no value to the reader and instead creates more confusion. e.g., what is "reaching a low inflammation level?"

Comments on the Quality of English Language

n/a